# Atomic structure evolution related to the Invar effect in Fe-based bulk metallic glasses

Alexander Firlus [1✉], Mihai Stoica [1], Stefan Michalik [2], Robin E. Schäublin[1] & Jörg F. Löffler [1✉]

The Invar effect is universally observed in Fe-based bulk metallic glasses. However, there is limited understanding on how this effect manifests at the atomic scale. Here, we use in-situ synchrotron-based high-energy X-ray diffraction to study the structural transformations of $(Fe_{71.2}B_{24}Y_{4.8})_{96}Nb_4$ and $(Fe_{73.2}B_{22}Y_{4.8})_{95}Mo_5$ bulk metallic glasses around the Curie temperature to understand the Invar effect they exhibit. The first two diffraction peaks shift in accordance with the macroscopically measured thermal expansion, which reveals the Invar effect. Additionally, the nearest-neighbor Fe–Fe pair distance correlates well with the macroscopic thermal expansion. In-situ X-ray diffraction is thus able to elucidate the Invar effect in Fe-based metallic glasses at the atomic scale. Here, we find that the Invar effect is not just a macroscopic effect but has a clear atomistic equivalent in the average Fe–Fe pair distance and also shows itself in higher-order atomic shells composed of multiple atom species.

[1] Laboratory of Metal Physics and Technology, Department of Materials, ETH Zurich, 8093 Zurich, Switzerland. [2] Diamond Light Source Ltd., Harwell Science and Innovation Campus, Didcot, Oxfordshire OX11 0DE, UK. ✉email: alexander.firlus@mat.ethz.ch; joerg.loeffler@mat.ethz.ch

Bulk metallic glasses (BMGs) possess exceptional mechanical (e.g., high yield strength and hardness)[1–4] and magnetic properties (e.g., high susceptibility and low coercivity)[3–5] due to their amorphous atomic arrangement. Since there is no long-range order (LRO) in metallic glasses, the short- and medium-range orders (SRO and MRO), resulting from the constituents' bonding preference, are the ones determining the material properties. One phenomenon that is strongly linked to SRO and MRO is thermal expansion. Most materials expand with increasing temperature, oftentimes at a constant rate. The rate of this expansion is defined by the relative volume change

$$\frac{V(T) - V_0}{V_0} = 3\alpha_1\big(T - T_0\big), \qquad (1)$$

where $V(T)$ and $V_0$ are the material's volume at temperatures $T$ and $T_0$, respectively, and $\alpha_1$ is the linear coefficient of thermal expansion (CTE).

In 1897 Charles Édouard Guillaume discovered that FeNi alloys with 36 wt% Ni have an exceptionally low CTE below the Curie temperature[6]. Around the Curie temperature, the CTE of the alloys increases drastically by up to one order of magnitude. This correlation of thermal expansion with the alloy's magnetic state became known as the Invar effect. Although this effect is rare in crystalline materials, it is universally observed in all ferromagnetic Fe-based BMGs[7–9]. It is worth pointing out that Fe–Ni increases its CTE by a factor of 10 at its dilatometric transition temperature, whereas the CTE of ferromagnetic Fe-based BMGs is only reduced by a factor of 2–5 in the ferromagnetic state compared to the CTE in the paramagnetic state (which is also close to that of pure Fe). Generally, BMGs have a constant (with respect to temperature) CTE, which is close to that of its main constituent elements[10]. Furthermore, it is interesting to note that crystalline alloys with the same bulk chemical composition as the Fe-based BMGs do not show the Invar effect, as illustrated in Supplementary Fig. 1. This suggests that it is the disordered atomic arrangement that gives rise to the Invar effect in BMGs, and that it is a combination of the short-ranged atomic potentials and the long-ranged magnetic interaction that enables the anomalously low CTE below their Curie temperature. Here we make a clear distinction between the dilatometric transition temperature at which structural changes happen and the Curie temperature at which the transition from ferromagnetism to paramagnetism occurs. Nevertheless, it is widely understood that the dilatometric transition is a magnetically driven effect and the two are closely related.

While it has been recognized that the Invar effect is universal in all ferromagnetic Fe-based BMGs[7–9], only little is known about the origins of this effect at the atomic level, because most of the experimental research has focused so far on its macroscopic manifestations. Some calculations and simulations with respect to the Invar effect in FeNi alloys[11] are based on a crystallographic unit cell that is of no use for amorphous materials. So far, literature reports observations of the Invar effect for a wide variety of Fe-based BMGs, but there is no framework on how it operates in amorphous materials. In the present work, we investigated the atomic-scale manifestation of the Invar effect to lay grounds for its understanding in BMGs.

Understanding the structure of metallic glasses is difficult due to their disordered atomic arrangement. One way to study structural rearrangements with increasing temperature is through molecular dynamics simulations[12]. These have been successfully employed in the past but they inherently rely on interatomic potentials to be known and are thus limited in the choice of alloys they can be applied to. Another way is to investigate atomic arrangements experimentally through X-ray absorption or scattering techniques.

There is one observation that strongly links the Invar effect in BMGs to Fe: it has been noted that a reduction of the Fe content or its partial replacement by another ferromagnetic element such as Ni reduces the Invar effect[7]. Moreover, it seems that the SRO and MRO also play a vital role in its strength. Heat treatments and relaxation close to the glass transition temperature are known to strengthen it, and it was noted that the melt temperature at casting can also influence its manifestation in BMGs[8].

While X-ray diffraction (XRD) is often used to study the atomic arrangement of BMGs, to this date there is no investigation of their atomic arrangement in relation to the Invar effect. Only two publications report the possibility of observing the Curie temperature in XRD experiments, exemplary on FeMnSiCuNbB[13] and FeCuNbMoSiB[14] BMGs, which is only possible because of the Invar effect. The large number of elements in the alloy complicates the extraction of pair distribution functions from the XRD experiments.

For this reason, here we investigate in this study quaternary alloy systems, i.e., $(Fe_{71.2}B_{24}Y_{4.8})_{96}Nb_4$ (denoted here QNb)[5,15,16] and $(Fe_{73.2}B_{22}Y_{4.8})_{95}Mo_5$ (further denoted QMo)[16]. Both have Fe as their only (ferro)magnetic element and possess good glass-forming ability. We perform time-resolved XRD and macroscopic measurements of the thermal expansion and magnetic properties to obtain information from the atomic to the macroscopic scale. From the XRD data, we also derive the pair distribution function of both alloys, which allows us to correlate the atomic structure with the Invar effect in these BMGs. We further show that the Invar effect is not just a macroscopic effect but has a clear atomistic equivalent in the average Fe–Fe pair distance and is seen in all higher-order atomic shells.

## Results

**X-ray diffraction.** Figure 1a, b displays the typical integrated diffraction profile. The XRD data confirm the amorphous character of both samples in their as-prepared state. The most apparent consequence of the heating is a reduction in the intensity of all diffraction peaks, in particular of the first one. There are five obvious diffraction halos of which the third one is overlapping with the second.

In order to quantify the effect of temperature changes on the atomic arrangement, the diffraction peaks were fitted with a Lorentzian function

$$L(x) = \frac{A}{2\pi}\frac{\Gamma}{(x - x_0)^2 + (1/2\Gamma)^2}, \qquad (2)$$

where $A$, $\Gamma$ and $x_0$ are the area, width and central position of the Lorentzian peak, respectively.

The peak positions are shown in Fig. 1c, d. The alloys were heated to their glass transition temperature (first heating), then cooled back to 323 K (cooling), and finally heated again beyond the glass transition temperature (second heating). We observe a shift of the peak positions to lower $q$ values as the temperature is increased, which corresponds to an expansion of the alloy. The rate of this shift is increasing at the dilatometric transition temperature and is marked with arrows in the figure. The peak ratio $\frac{q_2}{q_1}$ sharply increases at both the glass transition and the crystallization temperature, as illustrated in the insets. This means that the first peak shifts faster to lower $q$ than the second peak, which in turn illustrates that there is substantial atomic rearrangement at the length scale of the MRO. This figure also indicates that the QMo alloy was heated into the supercooled liquid region during the first heating and major changes to the atomic structure occurred, as can be seen from the substantial irreversible shift of the diffraction peak positions after the initial heating. However, the alloy still remained glassy. Apart from

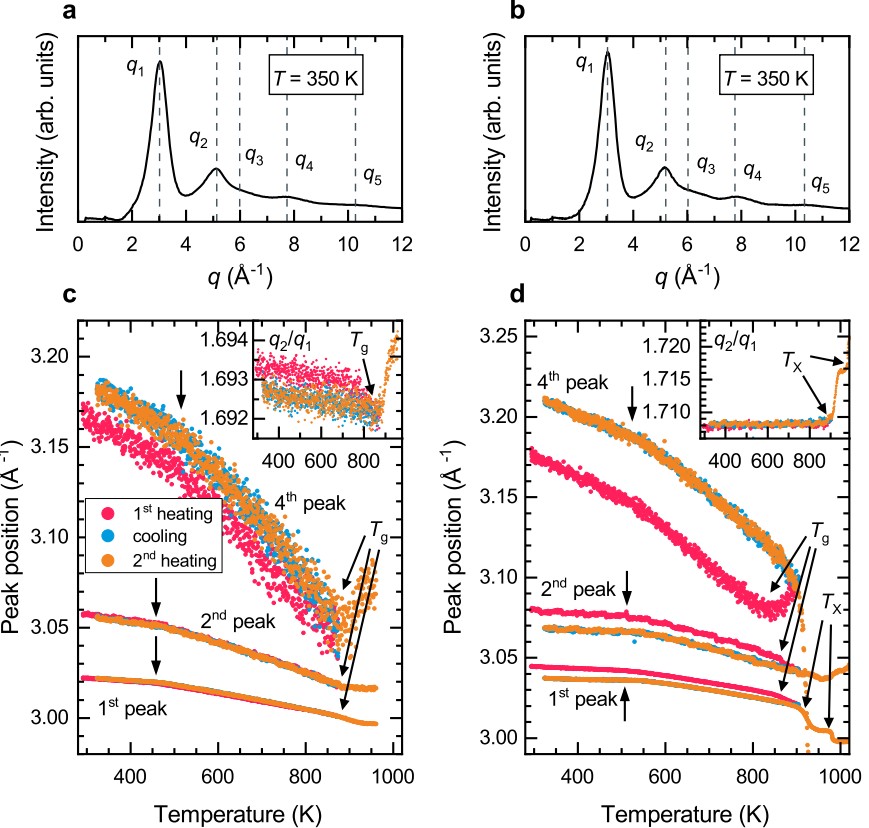

**Fig. 1 XRD radial intensity profiles of QNb and QMo and their temperature evolution.** After azimuthal integration of the 2D diffraction images, the 1D diffraction patterns display five broad peaks for both **a** QNb and **b** QMo. All peaks of **c** QNb and **d** QMo shift to lower $q$ values as temperature increases. A change in the shift rate is observed at the dilatometric transition temperature and marked with arrows. At temperatures far below the glass transition temperature, the changes are reversible. After the first heating run, there is a rearrangement of the SRO, which is reflected in the fourth peak position. Each inset shows the position ratio of the first two peaks as a function of temperature.

relaxation processes close to the glass transition temperature, the transition at the dilatometric transition temperature is fully reversible. The first heating changed the atomic structure close to the glass transition, while the cooling and second heating yield identical diffraction profiles.

**Dilatometry.** As temperature increases the material is expected to expand. This is reflected in a contraction of the diffuse diffraction rings. From the relative position changes of the broad diffraction peaks, it is possible to derive the relative volume change using the Yavari approach[17]

$$\left(\frac{q_0}{q(T)}\right)^3 = \frac{V(T)}{V_0},\qquad(3)$$

where $\frac{q(T)}{q_0}$ is the position of a peak in the diffraction profile relative to an arbitrary reference point and $\frac{V(T)}{V_0}$ is the relative volume change. By combining the Yavari approach with the definition of the CTE it is possible to obtain the CTE from the diffraction profile. While in principle the position of any peak can be used for calculating the relative volume change, only the first peak, corresponding to large interatomic distances, is expected to match the macroscopic behavior[18]. Higher-order peaks will also include effects of structural relaxation as glasses are metastable. Additionally, it is important to point out that the diffraction peaks in metallic glasses can develop independently because SRO and MRO can respond to temperature increases independently

and even in opposite ways[19,20]. This is contrary to crystalline materials where the diffraction peak positions have a fixed ratio.

We used the three prominent peaks $q_1$, $q_2$, and $q_4$ to derive the relative volume change from the XRD profile. Figure 2 shows the relative volume changes for both alloys as derived from these peaks, together with the macroscopically measured thermal expansion. Note that the volume changes from the first diffraction peak are supposed to represent the macroscopic behavior of the alloys[17,20]. The second and fourth diffraction peaks are associated with the nearest-neighbor atomic arrangement. The volume changes calculated using the shift of the higher-order diffraction peaks in Fig. 2 are offset for better readability. One can see that the QNb alloy loses some free volume when heated close to the glass transition temperature (see Fig. 2a). The end temperature of the first heating section was close to but did not extend beyond the glass transition temperature, which itself is clearly visible by the increase in thermal expansion of the first diffraction peak at high temperature. The QMo alloy (see Fig. 2d) on the other hand was heated into the supercooled liquid region during the first heating section, but according to the XRD patterns did not crystallize. As a consequence, a substantially different glassy state was created for the cooling and second heating section. This also becomes apparent through the large volume increase from the first heating to the cooling/second-heating section.

Linear fits of the CTEs with 99% confidence intervals were performed in the low-temperature and high-temperature regions. The CTEs are summarized in Table 1. The dilatometric transition temperature is taken to be the temperature at which the data diverges from the low-temperature linear fit and is marked in

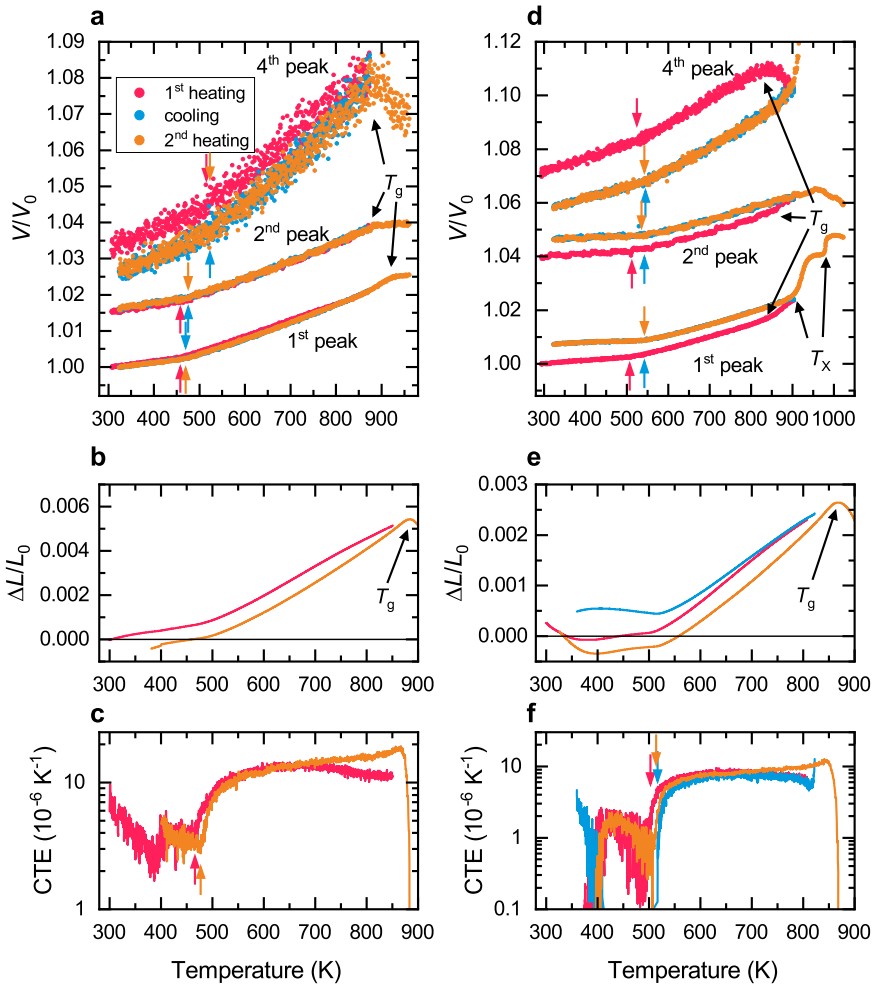

**Fig. 2 Relative volume expansion of QNb and QMo derived from the XRD patterns and macroscopic dilatometry.** The volume derived from XRD for **a** QNb and **d** QMo is offset for better visibility. The **b** length and **c** CTE of QNb shows the same expansion behavior as the XRD-derived volume change. The same applies to the **e** length and **f** CTE of QMo. The temperature at which the CTE increases is marked by arrows.

**Table 1 Coefficient of thermal expansion of QNb and QMo measured by XRD and dilatometry (DIL).**

**QNb**

| Section | First peak | | Second peak | | Fourth peak | | DIL | |
|---|---|---|---|---|---|---|---|---|
| | $\alpha_f$ [$10^{-6}$ K$^{-1}$] | $\alpha_p$ [$10^{-6}$ K$^{-1}$] | $\alpha_f$ [$10^{-6}$ K$^{-1}$] | $\alpha_p$ [$10^{-6}$ K$^{-1}$] | $\alpha_f$ [$10^{-6}$ K$^{-1}$] | $\alpha_p$ [$10^{-6}$ K$^{-1}$] | $\alpha_f$ [$10^{-6}$ K$^{-1}$] | $\alpha_p$ [$10^{-6}$ K$^{-1}$] |
| First heating | 5.4–6.0 | 14.8–14.9 | 5.0–6.6 | 15.3–15.7 | 13.5–22.0 | 33.2–37.0 | 3.6–5.4 | 11.4–14.7 |
| Cooling | 5.4–5.9 | 15.1–15.4 | 6.2–7.5 | 15.7–16.1 | 14.2–21.4 | 34.3–37.8 | - | - |
| Second heating | 4.8–5.4 | 15.1–15.3 | 5.5–6.9 | 15.3–15.8 | 13.6–20.2 | 33.6–36.9 | 3.4–5.0 | 11.1–17.0 |

**QMo**

| Section | First peak | | Second peak | | Fourth peak | | DIL | |
|---|---|---|---|---|---|---|---|---|
| | $\alpha_f$ [$10^{-6}$ K$^{-1}$] | $\alpha_p$ [$10^{-6}$ K$^{-1}$] | $\alpha_f$ [$10^{-6}$ K$^{-1}$] | $\alpha_p$ [$10^{-6}$ K$^{-1}$] | $\alpha_f$ [$10^{-6}$ K$^{-1}$] | $\alpha_p$ [$10^{-6}$ K$^{-1}$] | $\alpha_f$ [$10^{-6}$ K$^{-1}$] | $\alpha_p$ [$10^{-6}$ K$^{-1}$] |
| First heating | 3.5–4.0 | 14.0–14.3 | 2.7–4.4 | 13.2–13.8 | 13.9–17.6 | 29.8 - 31.2 | 0.6–1.7 | 7.0- 8.5 |
| Cooling | 2.5–3.1 | 12.9–13.2 | 2.0–3.7 | 11.7–12.5 | 12.7–15.3 | 27.3–28.5 | - | - |
| Second heating | 2.7–3.3 | 12.8–13.2 | 2.3–4.1 | 12.5–13.1 | 13.9–16.6 | 27.1–28.4 | 0.6–1.7 | 7.1–9.4 |

$\alpha_f$ is the CTE in the ferromagnetic state and $\alpha_p$ is the CTE in the paramagnetic state.

Fig. 2 with arrows. In all curves, one can identify the dilatometric transition temperature and glass transition temperature. The QNb alloy has a dilatometric transition temperature of approximately 458 K in the as-cast state. After relaxing the structure by heating close to $T_g$ the dilatometric transition temperature increases by about 12 K. The QMo alloy has a higher dilatometric transition temperature at 507 K in the as-cast state and increases to 542 K in the cooling and second heating section. It is interesting to note that the CTEs derived from the fourth peak for both alloys are significantly higher than those derived from the first

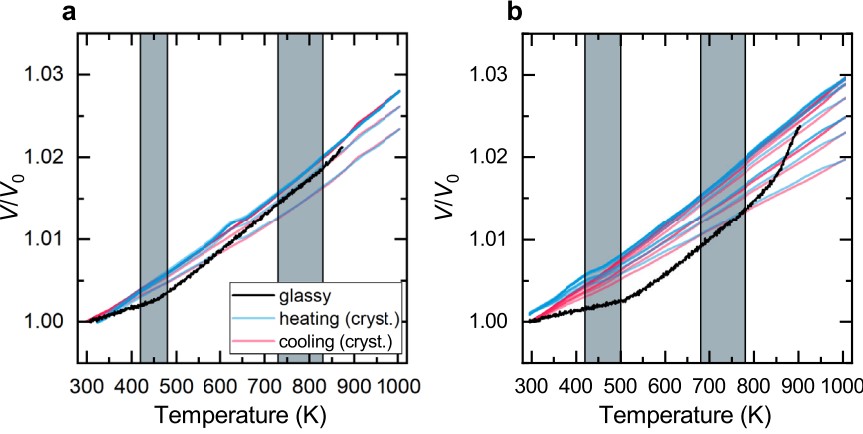

**Fig. 3 Relative volume change obtained from the diffraction peaks in crystallized QNb and QMo.** In the crystalline state, the alloys **a** QNb and **b** QMo do not show the Invar effect anymore. The volume change of the glassy alloys revealing the Invar effect is shown in black as a reference. The temperature intervals of the magnetic phase transitions of the crystalline alloys as obtained from magnetometry (see below) are shaded in gray.

two peaks. The QNb alloy additionally shows a much higher dilatometric transition temperature in its fourth diffraction peak position. Both the dilatometric transition temperature and the CTEs obtained from the first two diffraction peaks are in good agreement with the macroscopic dilatometry results, shown in Fig. 2b, c, e, f. For one, this can be explained by the fact that in XRD the CTEs are derived from a single diffraction peak and thus only consider part of the full structure. It was shown that only the first diffraction maximum corresponds to macroscopic volume changes, while higher-order diffraction peaks contain mostly information about a very local structure such as nearest-neighbor atoms[20]. Additionally, the Yavari approach is based on $V \sim L^3$ between the volume $V$ and length $L$, which comes from the Ehrenfest equation, but there is debate on whether that relationship also holds true for amorphous materials[19,21–23]. Without a crystallographic unit cell and no translational symmetry, different power-law relations between volume and length have been proposed with an exponent of around 2.3 to 2.5[21,22,24]. Applying this reduced exponent in the Yavari approach yields a lower CTE that is closer to the macroscopically measured one. However, the breakdown of the Ehrenfest equation is mostly relevant to temperatures above the glass transition temperature or even close to the liquidus temperature[19,21]. Since we focus on changes around the Curie temperature ($T_C \approx 450 - 550$ K), all temperatures involved in the analysis are well below the glass transition temperature ($T_g \approx 870$ K). Regardless of this, as far as the Invar effect is concerned, the Yavari approach allows not only to determine the dilatometric transition temperature from XRD experiments (via the Invar effect), as first shown by Michalik et al.[13], but also to quantitatively derive the CTE from the diffraction peak positions.

The Invar effect disappears once the samples are fully crystallized. The macroscopic dilatometry of the fully crystallized QNb and QMo alloys is shown in Supplementary Fig. 1 and reveals that the Invar effect is only present in the glassy samples. After crystallization, the CTE is roughly equal to that of the glassy samples above their Curie temperature (which is also the CTE of pure Fe). Figure 3 shows the relative volume change obtained from XRD of the crystallized QNb and QMo alloys. Neither alloy shows the Invar effect anymore. This clearly shows that it is a consequence of the disordered structure and the particular local atomic environment. Crystallizing the alloys makes the Invar effect disappear at the macroscopic and atomic length scale.

**Magnetometry.** In order to confirm that the dilatometric transition temperature is indeed related to a magnetic transformation

and not a purely structural one, we performed vibrating sample magnetometry (VSM) measurements. Figure 4 shows for both alloys the thermomagnetic curves and hysteresis at 300 K. The temperature profile is identical to the one used for the XRD experiments. After the first heating section, the Curie temperature and saturation of both alloys increase, as is commonly observed in Fe-based BMGs[9]. The cooling and second-heating thermomagnetic curves are exactly the same, which is well in agreement with the structural investigations by dilatometry and XRD. Compared to the XRD experiments, the second heating was extended up to 1273 K where the samples (partially) crystallized. From the second cooling thermomagnetic curves of both alloys, it is apparent that at least two ferromagnetic phases with different Curie temperatures are present after heating to 1273 K. One of those phases has a Curie temperature between 710 and 730 K and possibly results from a crystalline Fe-based phase. We could identify $Fe_{23}B_6$ and $Fe_{62}B_{14}Y_3$ in the crystallized alloys. FeB-based alloys with around 80 at.% Fe have indeed Curie temperatures that are around 700 K[25,26]. The other ferromagnetic phase has a magnetic phase transition close to the Curie temperature of the glassy state. This is probably the remains of the glassy phase that did not fully crystallize due to a mismatch in the glassy stoichiometry and the possible crystalline phases.

Additionally, it is observed that the saturation magnetization of the QMo alloy is much lower after crystallization than in the glassy state. This can be explained by non-ferromagnetic crystallization products. These may be antiferromagnetic Fe-rich clusters, paramagnetic fcc-like Fe-clusters, or B-rich amorphous FeB remains, which are known to have a Curie temperature below 300 K if their boron content is above 65 at.%[26].

When comparing the dilatometric transition temperature to the Curie temperature in Table 2, one can see that the dilatometric transition temperature (obtained from the first diffraction peak) is systematically larger than the magnetic one. This has also been observed to some extent by Michalik et al.[13] and Bednarcik et al.[14] in FeMnSiCuNbB and FeCuNbMoSiB glasses, respectively. However, the differences between the dilatometric transition temperature and Curie temperature are mostly within the error margin but consistently point in the same direction. Moreover, upon heat treatment, it appears that both the dilatometric transition temperature and the Curie temperature increase by about the same amount. The dilatometry experiments in combination with the thermomagnetic study thus show that both the dilatometric transition temperature and Curie temperature move in unison when subjected to a heat treatment. This

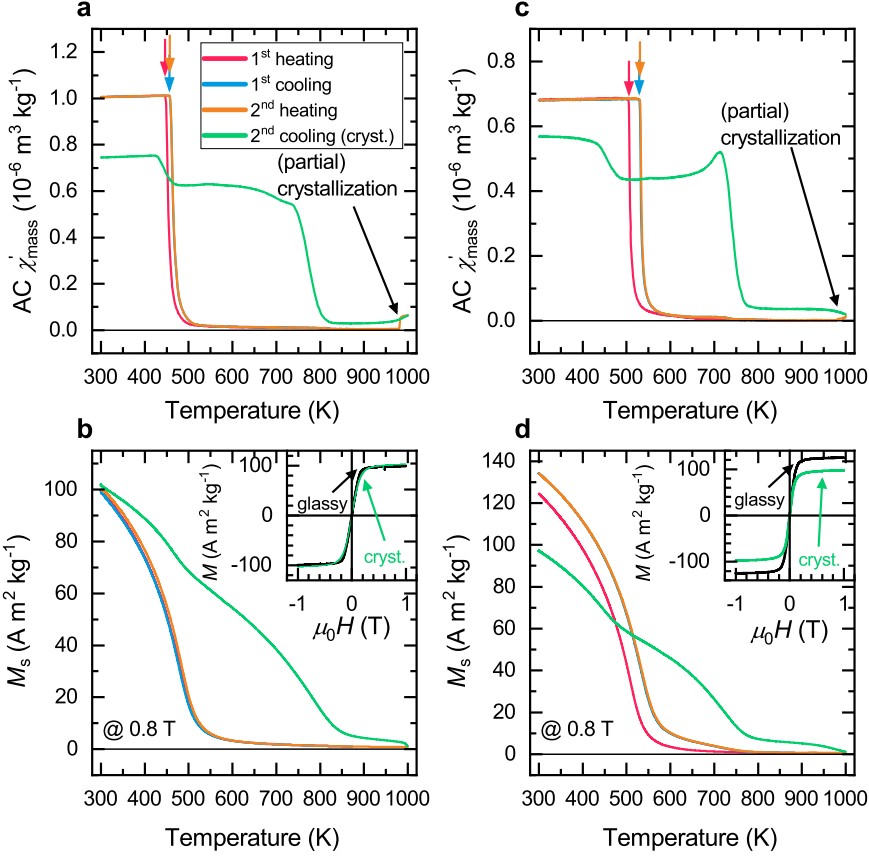

**Fig. 4 Thermomagnetic curves and magnetic hysteresis of QNb and QMo.** The AC susceptibility of **a** QNb and **c** QMo shows a clear transition from the ferromagnetic to the paramagnetic state (marked with arrows). The saturation magnetization as a function of temperature is shown in **b** for QNb and **d** for QMo. The hysteresis curves at 300 K for QNb and QMo are shown in the insets.

indicates that for the increased rate of thermal expansion to happen, all ferromagnetic coupling needs to vanish. In light of the universality of the Invar effect in Fe-based BMGs, this is not surprising as apparently any ferromagnetic glassy structure in these alloys is capable of lowering the CTE. This means that thermal excitations need to be large enough to destroy all ferromagnetic interaction before the Invar effect vanishes.

It is well known for the crystalline Invar alloy[11] and also believed to be true for metallic glasses[7,8] that the Invar effect is facilitated by Fe–Fe pairs. After the heating section, we observe an increase in the saturation magnetization and strengthening of the Invar effect, i.e., a further reduction of the CTE in the ferromagnetic state. After annealing, internal stresses induced by the rapid cooling are released. This usually comes with a reduction of coercivity due to the removal of stress-induced magnetic anisotropy. Consequently, more Fe atoms become free to participate in the ferromagnetic interaction and thus contribute to the Invar effect. This is observable as an increase in the saturation magnetization in the second run (see Fig. 4b, d). In the following, we look into the atomic arrangement to gain insights into how the Invar effect in amorphous materials works at the atomic scale.

**Pair distribution function.** Reduced pair distribution functions (rPDFs) were derived from the radial diffraction intensity profiles with the pdfgetx3 software[27] to gain insights into the atomic arrangement of the BMGs. A lower and upper integration limit of $q_{min} = 1.42\,\text{Å}^{-1}$ and $q_{max} = 15.60\,\text{Å}^{-1}$ was chosen for QNb, while for QMo the limits were $q_{min} = 1.48\,\text{Å}^{-1}$ and

$q_{max} = 14.30\,\text{Å}^{-1}$. Figure 5a, c shows the typical rPDFs for both alloys. In both alloys, there are five observable maxima, which represent atomic shells. The second shell is split into two sub-shells, which is a common observation in metallic glasses[3,28,29].

A first check of the quality of the rPDF as well as its relation to the glass transition and possibly the crystallization temperature can be obtained by investigating the total coordination number

$$\text{CN} = \int_{r_0}^{r_1} \left(4\pi r^2 \rho_{at} + rG(r)\right)dr, \tag{4}$$

where $\rho_{at}$ is the atomic density, $G(r)$ the reduced pair distribution function, and $r_0$ and $r_1$ the limits of the first atomic shell. The limits $r_0$ and $r_1$ were chosen in such a way that they are at the local minima of the integrands at room temperature. They were then kept fixed for all temperatures. The density was set as $\rho = 0.991 \sum_i f_i \rho_i$, where $f_i$ and $\rho_i$ are the weight concentration and density of element $i$. The prefactor 0.991 takes the disordered atomic structure and free volume into account. Thus we obtain atomic densities of $\rho_{at}^{QNb} = 90.6295\,\text{nm}^{-3}$ and $\rho_{at}^{QMo} = 90.6183\,\text{nm}^{-3}$.

For a perfectly icosahedral SRO the coordination number would be $\text{CN}_{ico} = 12$. We observed a higher coordination number very close to $\text{CN}_{obs} = 14.0$ for both alloys. While this is significantly larger, it has been found through XRD, neutron diffraction, and extended X-ray absorption fine structure (EXAFS) experiments in combination with reverse Monte Carlo simulations that ternary FeBNb glasses have a CN $\approx 14$[25], especially when disregarding boron, which is mostly transparent to X-rays. This holds for FeBNb glasses with various chemical compositions. As the ternary FeBNb system forms the basis of our QNb alloy, it is reasonable to assume

**Table 2 Curie temperature and dilatometric transition temperature of QNb and QMo measured by XRD peaks (1, 2, and 4), dilatometry (DIL) and magnetometry (AC).**

**QNb**

| Section | XRD1 (±10 K) | XRD2 (±10 K) | XRD4 (±10 K) | DIL (±5 K) | AC (±5 K) |
|---|---|---|---|---|---|
| First heating | 458 | 458 | 515 | 466 | 446 |
| Cooling | 470 | 475 | 523 | - | 456 |
| Second heating | 470 | 475 | 523 | 477 | 457 |

**QMo**

| Section | XRD1 (±10 K) | XRD2 (±10 K) | XRD4 (±10 K) | DIL (±5 K) | AC (±5 K) |
|---|---|---|---|---|---|
| First heating | 507 | 512 | 524 | 503 | 505 |
| Cooling | 542 | 542 | 543 | 517 | 529 |
| Second heating | 542 | 542 | 543 | 514 | 530 |

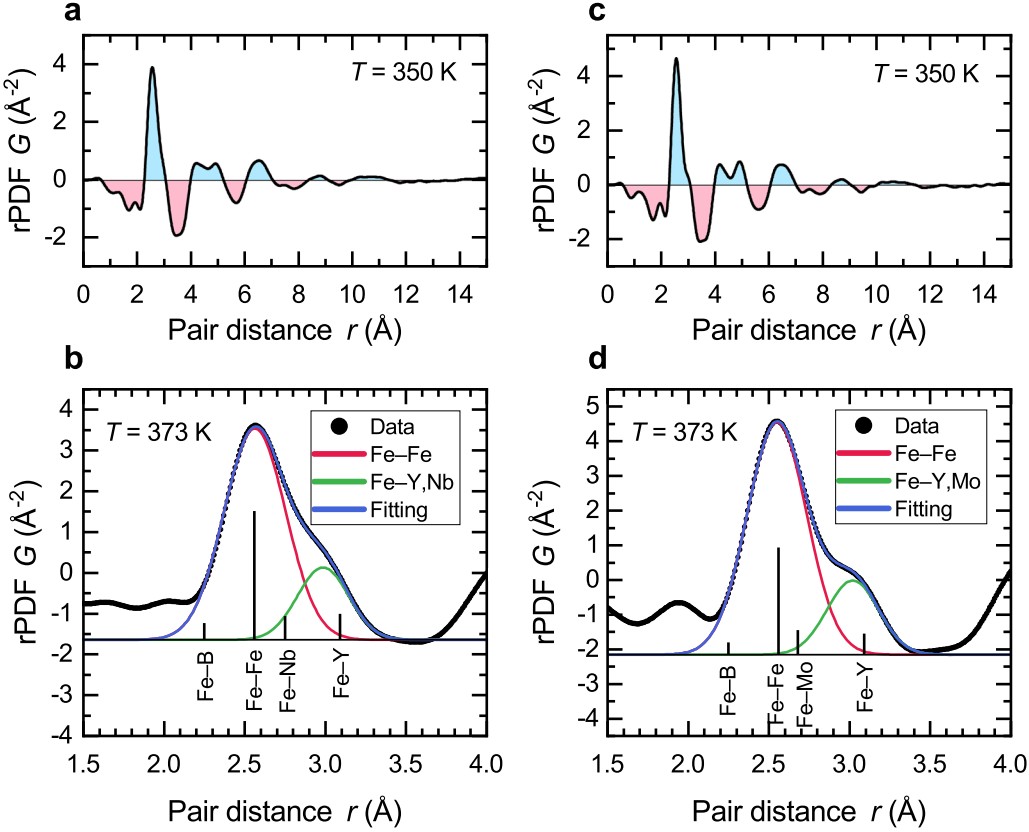

**Fig. 5 Reduced pair distribution function of QNb and QMo.** The rPDF shows five atomic shells for both **a** QNb and **c** QMo. The first atomic shell of **b** QNb and **d** QMo can be decomposed into two Gaussian distributions. The first atomic shell is dominated by Fe–Fe pairs. Fe–Y and Fe–Nb/Mo pairs form a common distance distribution.

that its structure is similar. The same holds true for the QMo system which, compared to QNb, hosts Mo instead of Nb. In terms of improving the glass-forming ability and bonding behavior, both elements have been reported to behave similarly in BMGs[16]. Therefore, we can consider the rPDF to be of sufficient quality for quantitative analysis.

In order to investigate the changes in the rPDF further, we decomposed the first atomic shell into its constituent pairs. For this, it was fitted with a sum of Gaussian functions

$$G(x) = \frac{A}{\sqrt{2\pi\sigma^2}} \exp\left(-\frac{1}{2}\left(\frac{x-\mu}{\sigma}\right)^2\right), \quad (5)$$

where $A$, $\mu$, and $\sigma$ are the area, mean, and standard deviation of

the distribution, respectively. An example of the fitting is shown in Fig. 5b, d. The first atomic shell is characterized by the main peak with a shoulder that extends to higher atomic distances for both alloys. This shoulder is more pronounced for the QMo alloy than for QNb. To identify the atomic pairs responsible for the features of the rPDF, we calculated the X-ray weighting factors[30], $w_{ij}$, for all atomic pairs by

$$w_{ij} = \frac{1}{\left(\sum_k c_k f_k\right)^2} \begin{cases} c_i^2 f_i^2, & i = j \\ 2c_i c_j f_i f_j, & i \neq j \end{cases} \quad (6)$$

with $c_k$ the atomic concentration of element $k$ and $f_k$ the X-ray scattering factor at $q = 0$ of element $k$ at the used wavelength

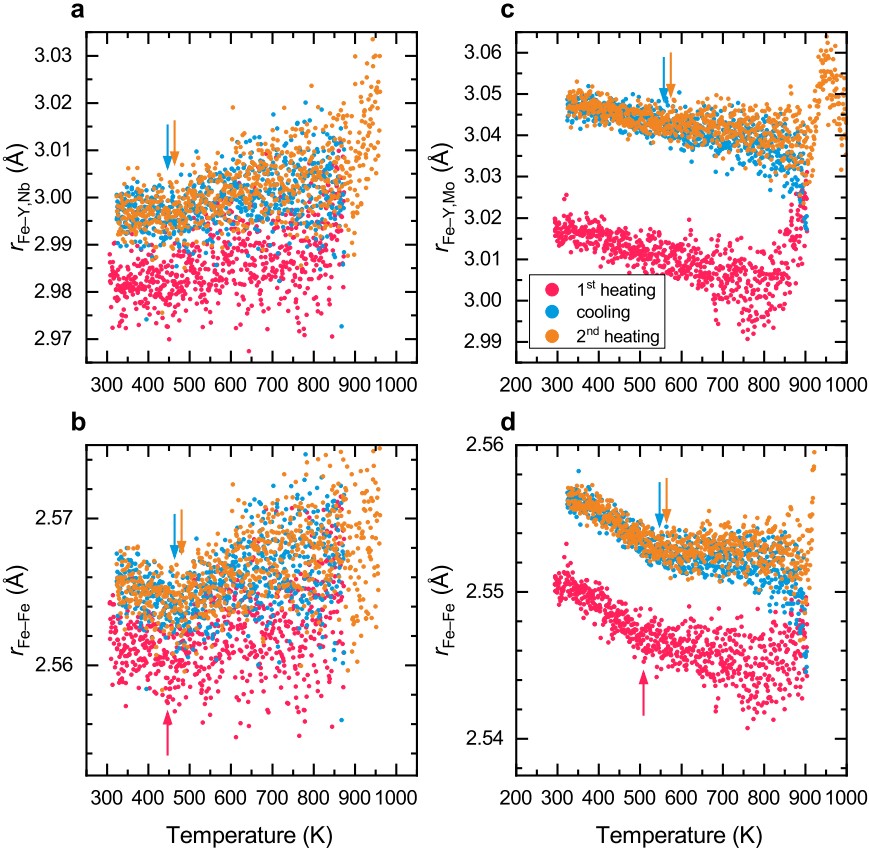

**Fig. 6 Average atomic pair distance of the main constituents in the first atomic shell of QNb and QMo.** The first atomic shell of QNb consists of **a** Fe–Y,Nb and **b** Fe–Fe in QNb. In QMo, the first atomic shell is comprised of **c** Fe–Y,Mo and **d** Fe–Fe. The average Fe–Fe distance is reduced at the dilatometric transition temperature (marked with arrows) and a change in the contraction rate is clearly visible. This effect is stronger after the sample has been heated close to the glass transition temperature in the first heating run. After heat treatment also the Fe–Y,Nb and Fe–Y,Mo atomic pairs show a change in the expansion rate at the dilatometric transition temperature.

(taken from Chantler[31]). These weighting factors, which contribute at least 3%, are marked in Fig. 5 at the corresponding atomic pair distances based on the sum of the constituents' metallic or van-der-Waals radii, which are 1.28 Å, 0.97 Å, 1.81 Å, 1.47 Å, and 1.40 Å for Fe, B, Y, Nb, and Mo, respectively. While there are ten possible pairs, many of them are negligible because of the low scattering power of B and the low concentration of Y and Nb/Mo. The main contribution to the first atomic shell comes from Fe–Fe pairs in both alloys. This is to be expected since Fe is by far the most abundant element. The center position of the first Gaussian function is in excellent agreement with the metallic Fe–Fe distance. The position of the second center does not coincide with Fe–Y or Fe–Nb/Mo, which are the second and third most scattering atomic pairs. Instead, the center is located between the Fe–Y and Fe–Nb/Mo atomic pair distances. This leads to the suggestion that Y and Nb/Mo all bond with Fe in the same way and produce a single pair distribution instead of two separated distributions. In fact, while this similarity in bonding preference of Y and Nb has already been observed through a combination of EXAFS and computational means[32], we observe the same behavior also for Fe–Y/Mo bonds.

The evolution of the deconvolution parameters of both alloys is shown in Fig. 6. At low temperatures, the Fe–Fe distance in both alloys decreases as the temperature increases. The critical temperature where the Fe–Fe distance deviates from the linear relationship with respect to temperature is in good agreement with the dilatometric transition temperature. Above the dilatometric transition temperature, the Fe–Fe pairs contract less

rapidly or even reveal a local minimum. This observation is even clearer after the samples have been heated once close to $T_g$, which is in agreement with the previous observations of increased strength of the Invar effect after heat treatment. The low-temperature Fe–Fe contraction is small (around 0.1%) but nonetheless statistically significant. The observation of the glass transition temperature and the overlapping of the cooling and the second heating curves are clear indications that the experimental setup and analysis are consistent and sensitive enough to pick up the average atomic distance changes. Indeed, a linear fit of the Fe–Fe distance as a function of temperature up to $T_C$ estimates the slope to be 4 to 7 standard deviations away from 0. This clearly excludes noise as a possible source of the Fe–Fe contraction below $T_C$. A systemic error can also be excluded, given the fact that the change in the contraction rate is reproducible during all heating and cooling sections with the occurrence of the same critical temperature that also matches the dilatometric transition temperature. Thus, we see that the average Fe–Fe distance is linked to the magnetic state of the alloy, which in turn indicates that the Invar effect relates to changes in the electronic structure of the Fe atoms. In this case, there is no LRO on which an explanation could be based. This is supported by the fact that fully crystallized BMGs do not show any Invar effect anymore.

The Fe–Y,Nb and Fe–Y,Mo pairs appear to be also affected by the magnetic state of the sample because the average pair distance increases faster above the Curie temperature (see Fig. 6a, c). This can be seen mostly in the heat-treated samples. Without any

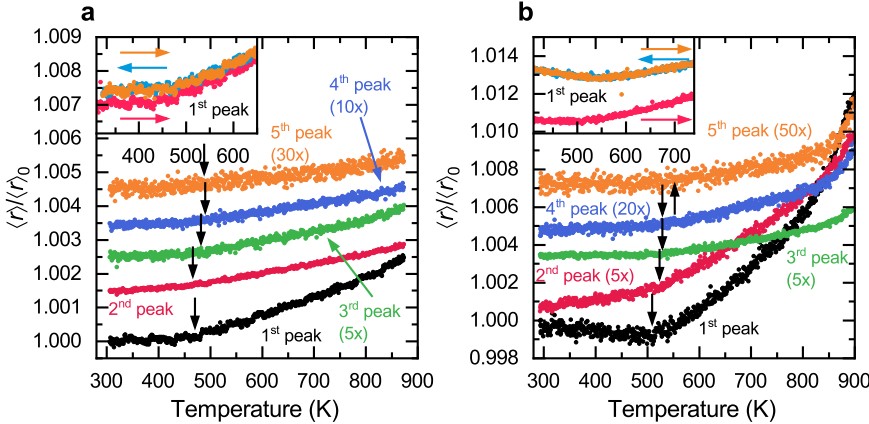

**Fig. 7 Increase of the average atomic distances for the various atomic shells in QNb and QMo.** All atomic shells of both **a** QNb and **b** QMo show signs of the Invar effect through a change of slope at the dilatometric transition temperature. This effect is particularly strong in the first atomic shell and gets stronger after relaxation. The higher-order peak positions are shifted for clarity. The same nominal increase of atomic distances becomes proportionally less significant at higher-order atomic shells; the relative changes of average atomic distances are thus scaled for better readability. The insets show the heating and cooling runs for the first atomic shell, where the heating runs differ from each other due to structural relaxation at the end of the first heating.

further information about the local atomic structure, however, it is hard to conclude whether the magnetically driven contraction affects the paramagnetic elements (i.e., Y, Nb, Mo) or whether the Fe network exerts pressure on the remaining atoms to facilitate their thermal expansion/contraction. If the Invar effect is caused on the atomic level by an element-selective contraction of only Fe–Fe bonds, only Fe-rich alloys can show the Invar effect because the energy gain from the magnetically induced contraction has to exceed the energy cost of the distortions of the other atom species. In fact, the classical FeNi Invar alloy is found on the Fe-rich side and for Fe-based BMGs it has been noted that the strength of the Invar effect, i.e., the difference in CTE before and after the dilatometric transition temperature, is very sensitive to the total Fe content[7].

Note again that the first heating section of QMo expanded into the supercooled liquid region and thus caused irreversible changes to the glassy structure. These changes appear particularly visible in the Fe–Y/Mo pair distribution.

Since the mean position of atomic pairs disregards changes to the width and shape of the individual atomic shells, we further looked into the central position of the whole atomic shells that includes all atom species and is defined as

$$\langle r \rangle = \frac{\int_{r_0}^{r_1} r \left( \frac{G(r)}{4\pi\rho_{at}r} + 1 \right) dr}{\int_{r_0}^{r_1} \left( \frac{G(r)}{4\pi\rho_{at}r} + 1 \right) dr}, \quad (7)$$

where $r_0$ and $r_1$ are the boundaries of any atomic shell in the rPDF $G(r)$. They were chosen such that a local minimum in $G(r)$ is found at both $r_0$ and $r_1$ at room temperature and they were kept fixed for all temperatures. The qualitative shape of $\langle r \rangle$ is independent of the precise integration limits and even the absolute value of $\langle r \rangle$ is very insensitive to the integration limits. A variation of the integration limits by up to 10% changes $\langle r \rangle$ by less than 3% and thus justifies the temperature-independent integration limits. Since the rPDF measures the atomic density in excess of the bulk density, $\langle r \rangle$ is a measure of the average atomic pair distance due to bonding preferences, and also takes the shape changes including widening into account. As seen in Fig. 5, the first atomic shell can be deconvoluted into two separate pair distance distributions. Both widen with increasing temperature, which results in an increase of $\langle r \rangle$ when the widening predominantly generates larger pair distances rather than shorter ones. Figure 7 shows the average pair distance of the atomic shells

as a function of temperature. The dilatometric transition temperature is clearly visible at $T_C \approx 460$ K for QNb and $T_C \approx 510$ K for QMo. The shift in Curie temperature is also apparent and in agreement with the dilatometric experiments (see Fig. 2). Below $T_C$ the average pair distance is almost constant or even decreases with increasing temperature for both alloys (see, in particular, the insets to Fig. 7) and the heat-treated alloys (cooling and second heating) show a correspondingly stronger change at $T_C$. At the same time, the dilatometry and XRD experiments reveal a decreased CTE in the ferromagnetic state and an increased strength of the Invar effect ($\triangle \alpha = \alpha_p - \alpha_f$) after relaxation (see Fig. 2). It is known from ab inito calculations[11] and EXAFS experiments[33] that in crystalline materials the Invar effect can be clearly attributed to Fe atoms, but for metallic glasses, we find that it is also visible in the full atomic shell, which comprises all atomic species.

Figure 7 also illustrates that the Invar effect is visible in the average position of higher-order atomic shells. For those atomic shells, there is no contraction below $T_C$ but the change in slope is still apparent. The average atomic distance of any shell is identical for the cooling and second-heating runs, but both differ from the first heating section due to structural relaxation at the end of the first heating. An example is shown in the insets of Fig. 7 for the first atomic shell. For the QMo alloy, this difference is particularly large because the alloy was heated beyond the glass transition temperature at the end of the first heating section, which led to a change in the glassy structure. The fact that the Invar effect can still be seen is further evidence that it is prevalent in all glassy states of this alloy.

## Discussion
$(Fe_{71.2}B_{24}Y_{4.8})_{96}Nb_4$ (QNb) and $(Fe_{73.2}B_{22}Y_{4.8})_{95}Mo_5$ (QMo) bulk metallic glasses were studied for their changes in atomic arrangement related to the Invar effect and Curie temperature. The Yavari approach[17] was used to derive the thermal expansion from the XRD intensity profile. We illustrated that this method is not only capable of determining the dilatometric transition temperature but also allows quantifying the coefficient of thermal expansion (CTE). Furthermore, the Invar effect is visible in all diffraction peaks, illustrating that it is present at multiple length scales in the atomic arrangement. The first atomic shell of both alloys was deconvoluted into the main contributing atomic pairs. It is dominated by Fe–Fe pairs and Fe–Y and Fe–Nb/Mo pairs,

the latter two belonging to a common Gaussian in the pair-distance distribution.

We find that the Invar effect in BMGs is not just a macroscopic effect but has a clearly observable atomistic origin. It is facilitated by the magnetic interaction of Fe–Fe. As temperature increases, the magnetic interaction counteracts the normal thermal expansion and can even result in a net contraction of Fe–Fe below the Curie temperature. This effect is fully reversible at the Curie temperature but can be strengthened by annealing the glasses at their glass transition temperature. Crystallization on the other hand completely destroys the Invar effect.

Additionally, also higher-order atomic shells show the Invar effect at the atomic level. The average distance of all atomic shells shows different expansion rates below and above the dilatometric transition temperature. These higher-order atomic shells cannot be attributed to specific atomic species, pairs, or arrangements but rather show that the Invar effect can be observed at all length scales from the interatomic distances to the macroscopic scale. The change in the average-distance expansion rate as a function of temperature is also fully reversible at the Curie temperature. Nevertheless, this change is most prominent in the first atomic shell, which suggests a strong link between SRO and the Invar effect. The observation of the Invar effect in the rPDF at all length scales also suggests that any disordered atomic arrangement contributes to it.

## Methods

**Sample preparation and characterization.** The quaternary (FeBY)Nb (QNb) and (FeBY)Mo (QMo) master alloys were prepared from pure elements with purities of 99.5% for B and at least 99.95% for Fe, Y, Nb, and Mo. The QNb alloy was produced from induction-melted FeY and FeNb eutectic alloys with the addition of pure Fe, B, and Y through arc melting in Ti-gettered argon (99.999% pure) atmosphere. The resulting button was flipped over and remelted several times to ensure homogeneity. The QMo alloy was prepared by alloying Fe and B via induction melting, and subsequent arc melting of the FeB alloy with Y and Mo. The samples were cast in the shape of rods with 2 or 3 mm diameter by pulling the molten alloys into water-cooled copper molds. Inductively coupled plasma optical emission spectrometry (ICP-OES) confirmed the nominal chemical compositions, which are summarized in Supplementary Table 1. Differential scanning calorimetry (DSC; NETZSCH DSC 404 C) was performed on both alloys (see Supplementary Fig. 2) and the thermophysical properties are in agreement with previous works[5,15,16]. In addition to the XRD experiments, the amorphous nature of the samples was confirmed by high-resolution transmission electron microscopy (HRTEM) measurements using an FEI Talos F200X. The HRTEM images are shown in Supplementary Fig. 3.

The crystalline samples were prepared from the amorphous samples by heating them twice to 1200 K and performing an isothermal heat treatment at this temperature for at least 60 min.

**Dilatometry.** Several millimeter-long sections of the rods were cut for the dilatometry experiments (DIL, Netzsch DIL 402 C), and the surfaces were polished to ensure parallel faces. A heating/cooling rate of 5 K min$^{-1}$ was used with 15–30 min isothermal sections between each change of heating/cooling direction in order to stabilize the temperature controller. Dilatometry was performed in a 99.999% pure argon atmosphere with an 80 ml/min flow rate. The temperature calibration was done by observing the melting points of pure In, Sn, Pb, Zn, eutectic AgCu, Ag, and Cu.

**Magnetometry.** The magnetic properties of the alloys were investigated by vibrating sample magnetometry (VSM) at 15.9 Hz with an MPMS3 device (QuantumDesign). The thermomagnetic curves were measured at saturation by applying a magnetic field of 0.8 T. The AC susceptibility (i.e., response to an alternating magnetic field) of the alloys was studied with the same device with an excitation of 400 A m$^{-1}$ at 113 Hz without any bias field applied. Errors in the measurement of the magnetic moments were confirmed to be less than 1% with a Pd reference sample and the temperature controller had an error of less than 5 K, which was confirmed by the Curie temperature of FeNi and pure Ni reference samples.

**High-energy X-ray diffraction.** The in-situ XRD experiments on the glassy samples were performed on the I12-JEEP beamline[34] at the Diamond Light Source (Didcot, United Kingdom) in transmission geometry using a monochromatic X-ray beam of energy 112.37 keV. The acquisition time per frame was 10 s with a beam size of 0.25 mm × 0.25 mm. The beam intensity was normalized with respect to the current in the storage ring. Sample disks were cut from the center of the rods and polished down to 0.5–0.8 mm thickness. The samples were then investigated in situ by XRD at a constant heating and cooling rate of 5 K min$^{-1}$ in a Linkam TS1500 heating stage. The inner heating-stage body was purged with argon gas. The heating profiles for both alloys consist of three sections: (i) heating from room temperature to the glass transition temperature, (ii) cooling back to 323 K, and (iii) second heating beyond the glass transition temperature. The XRD experiments on the crystalline samples were done similarly at 80.393 keV with a beam size of 0.25 mm × 0.25 mm and 10 s acquisition time per frame. The exact beam energy, the detector-to-sample distance, and the 2D Pilatus 2 M CdTe detector geometry (for example, the center of the beam position and the orthogonality of the detector) were calibrated by measuring a CeO$_2$ NIST standard at multiple detector positions[35]. The calibration procedure together with the data integration along the radius of the diffraction circles in (reciprocal) $q$ space was realized using the DAWN software[36].

## Data availability

All data are available from the corresponding authors on request.

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

## Acknowledgements

The authors thank C. Wegmann and S. Reitz for technical help and L. Duarte for support with the ICP-OES measurements. This work was carried out with the support of the Diamond Light Source, I12-JEEP beamline, proposal nt26608-1. The authors also express special thanks to Š. Stanko for his help with the measurements at I12-JEEP, and gratefully acknowledge the support by an ETH Research Grant (ETH-47 17-1; J.F.L.) and the ETH+ initiative within the framework of SynMatLab (Laboratory for Multiscale Materials Synthesis and Hands-On Education; J.F.L.). The authors also acknowledge the ETH Zurich Scientific Center for Optical and Electron Microscopy (ScopeM) for providing access to its instruments.

## Author contributions

J.F.L., M.S., and A.F. designed the study and J.F.L. supervised the work. A.F. prepared the samples and performed the dilatometry and magnetometry measurements. A.F., M.S., and S.M. planned the XRD experiments and A.F. and S.M. carried them out. A.F. and R.E.S. performed the HRTEM measurements. A.F. analyzed and correlated the data. All authors discussed and validated the results and wrote the manuscript.

## Competing interests

The authors declare no competing interests.
