## [Peer Review File · Nature Communications]

Title: Atomic structure evolution related to the Invar effect in Fe-based bulk metallic glassesREVIEWER COMMENTS

Reviewer #1 (Remarks to the Author):

In this paper, the authors use in-situ synchrotron-based high-energy X-ray diffraction to study the structural transformations of $(\text{Fe}_{71.2}\text{B}_{24}\text{Y}_{4.8})_{96}\text{Nb}_4$ and $(\text{Fe}_{73.2}\text{B}_{22}\text{Y}_{4.8})_{95}\text{Mo}_5$ bulk metallic glasses around the Curie temperature to understand the Invar effect they exhibit. It was found that combining atomic-scale information with macroscopic dilatometry and magnetometry experiments allows to bridge multiple orders of magnitude in length scale and improve the understanding of the Invar effect in metallic glasses at the atomic scale. The authors have done systematic and professional experiments, and gave very beautiful data support the conclusions. I think this paper is very interesting and worth considering to accept in the Nature Communications, after making some necessary revisions.

(1) In order to more clearly characterize the amorphous structure of the two samples, the DSC curve and high-resolution transmission electron microscope of the samples should be given.

(2) As we all know, the Invar effect is also widespread in crystalline materials. Then it is necessary to clarify the mechanism of the difference between the effect in the crystal and the Invar effect in the amorphous alloys.

(3) This paper mainly clarifies that the Invar effect is related to the distance between adjacent atoms between Fe-Fe. In fact, alloys with Invar effect are alloys with special compositions. It should be the Invar effect caused by the interaction of these components with Fe atoms. Can you predict through this work, what kind of alloy composition may have the Invar effect?

(4) Under normal circumstances, the saturation magnetization of Fe-based metallic glasses after annealing and crystallization is higher than that of the amorphous state. Why the saturation magnetization after crystallization in Figure 4(d) is much lower than that in the amorphous state. It should give a more systematic explanation in the paper.

(5) In order to better clarify the structural origins of the Invar effect, I think the LAMMPS simulations are necessary here.

Reviewer #2 (Remarks to the Author):

The paper contains interesting information and merits publication. However, some modifications are required to be made before it can be published.

Writing about Invar effect in these bulk metallic glassy alloys one shall keep in mind that the CTE of metallic glasses based on iron is only 1.5-2 times lower than the CTE value for pure iron. Compare it with 1.5×10^{-6} K for Invar which is an order of magnitude lower than that of pure Fe. However, it is indeed lower than that of the base component which makes it different from other BMGs [Scripta Materialia, 58, (2008), 1106-1109]. One can also mention Cu-Zr based alloys CTE of which ($\sim 13 \times 10^{-6}$ K) is close to the average of CTE found for Cu and Zr.

Did not the authors perform the same procedure for a crystallized BMG sample? One could have monitored the increase in the interatomic distances using Bragg peaks and compare with the BMG case.

Main complaint is that the manuscript does not contain a clear explanation of the nature of the observed effect. It shall be stated more clearly in the discussion section. I would recommend to perform ab-initio MD simulation of the atomic cells at several temperature values like it was done in [Appl. Phys. Lett. 93, 113104 (2008) and JOURNAL OF APPLIED PHYSICS 110, 043519 (2011)], for example.

Contribution of the magnetic effects was studied in [PHYSICAL REVIEW B 68, 014407, 2003].

One phrase should be reconsidered:

- "Expansion of the average atomic distances..." can distances expand?

Reviewer #3 (Remarks to the Author):

The anomalously low thermal expansion, Invar effect, is one of the oldest solid state physics phenomena, which still has not got a commonly accepted explanation. For metallic glass, the situation is more difficult since the origin of the local magnetic movement is still unclear, not to mention the magneto-volume effect varied with temperature. So far, whether in the crystalline state or in the glass state, theoretical or experimental, nearly all investigations on Invar effect in atomic level are at 0 K or with some specific compositions, in situ experiments along temperature are very few.

This work employs in-situ synchrotron-based high-energy X-ray diffraction to study the structural transformations of two Fe-based bulk metallic glasses around the Curie temperature. Combining atomic-scale information with macroscopic dilatometry and magnetometry experiments, it is found that the first two diffraction peaks shift in accordance with the macroscopically measured thermal expansion, and the nearest-neighbor Fe-Fe pair distance correlates well with the macroscopic thermal expansion. These are crucial to improve the understanding of the Invar effect in both metallic glasses and crystals at the atomic scale. So, this work is worth in publishing in Nature Communications.

Some revisions should be made.

1. The definition of the rate of volume expansion must be wrong, since at $T=T_0$, the equation is not equal.
2. In Figure 2b, why T_g is not observed in the curve of the 1st peak?
3. In general, the first sharp diffraction peak around 3.0 \AA^{-1} corresponds to short-range order, however, the authors ascribed the shift of this peak (In Figure 2) to medium-range order. Then, what is the length scale of the so-called MRO?
4. The equation to calculate the coordination number may be wrong with the '+' in the integral.
5. In calculating the weight factor of X-ray scattering, f_i is used with different meaning from the previous description. f_p is actually the X-ray scattering factor of element p at $q=0 \text{ \AA}^{-1}$.
6. In the calculation of $\langle r \rangle$, I think the integral limits are not correctly set. $r_0 \sim r_1$ should spans the first coordination shell.
7. "It is known that in crystalline materials the Invar effect can be clearly attributed to Fe10 atoms, but for metallic glasses we find that it is also visible in the full atomic shell, which comprises all atomic species." I remind the authors that some investigations focus on the chemical short-range order around Fe atoms in Fe-Ni crystalline Invar alloys.

Alexander Firlus
ETH Zurich
Vladimir-Prelog-Weg 1-5
Zurich, CH 8093
alexander.firlus@mat.ethz.ch
+41 44 633 63 62

[November 16, 2021]

Dear reviewers,

we would like to express our appreciation to all reviewers for their valuable comments on our manuscript. We are pleased to hear the reviewers found our manuscript *very interesting and worth considering to accept in the Nature Communications, contains interesting information and merits publication and is worth in publishing in Nature Communications*. Revisions in the manuscript are shown in red font. In the following we would like to comment on the reviewers' feedback point by point:

(1) Reviewer 1: *In order to more clearly characterize the amorphous structure of the two samples, the DSC curve and high-resolution transmission electron microscope of the samples should be given.*

Thank you for this suggestion. Initially, we did not include the DSC curves as they show no difference from what has been published already, but we agree that it confirms a good sample preparation and we therefore added them to the supplementary materials. Likewise, we added HR-TEM images to the supplementary materials to further confirm the amorphous nature of our samples.

(2) Reviewer 1: *As we all know, the Invar effect is also widespread in crystalline materials. Then it is necessary to clarify the mechanism of the difference between the effect in the crystal and the Invar effect in the amorphous alloys.*

Thank you for this comment. While it is true that the Invar effect is observed in more crystalline alloys than just $\text{Fe}_{64}\text{Ni}_{36}$, we would not consider it a widespread effect. The Invar effect exists only in a narrow compositional range for crystalline alloys and furthermore requires a specific local atomic environment. However, the Invar effect exists for all ferromagnetic BMGs irrespective of their Fe-content and the addition of other alloying elements. This is indicating that the disorder in the atomic arrangement is responsible for making the Invar effect so common in Fe-based bulk metallic glasses (Fe-BMGs). In this work we provide experimental evidence that the Invar effect in Fe-BMGs is not just a macroscopic effect but instead has clearly observable atomistic origins despite the lack in translational symmetry.

(3) Reviewer 1: *This paper mainly clarifies that the Invar effect is related to the distance between adjacent atoms between Fe-Fe. In fact, alloys with Invar effect are alloys with special compositions. It should be the Invar effect caused by the interaction of these components with Fe atoms. Can you predict through this work, what kind of alloy composition may have the Invar effect?*

We appreciate this suggestion very much. All ferromagnetic Fe-BMGs seem to show the Invar effect as has been shown in [Hu, Q. *et al.* Intermetallics 93, 318-322 (2018)]. Through our work we can attribute the atomic scale manifestation of the Invar effect to Fe–Fe bonds. In particular, we observe a Fe–Fe contraction below the Curie

temperature, while the alloy is still expanding on the macroscopic scale. This indicates that only Fe-rich alloys (or in general alloys rich in the "Invar"-active species) can show the Invar effect on a macroscopic scale, as there is a balance between the magnetically induced contraction of Fe-Fe and the deformation/expansion of all the other bonds. In fact, it has been observed that the strength of the Invar effect in Fe-BMGs is very sensitive to the total concentration of Fe. Additionally, the crystalline alloys with the Invar effect are found on the Fe-rich side of the phase diagrams. Nevertheless, a high Fe concentration is by far not sufficient for the Invar effect to manifest. We have added this prediction to the manuscript.

(4) Reviewer 1: *Under normal circumstances, the saturation magnetization of Fe-based metallic glasses after annealing and crystallization is higher than that of the amorphous state. Why the saturation magnetization after crystallization in Figure 4(d) is much lower than that in the amorphous state. It should give a more systematic explanation in the paper.*

Thank you for raising this concern. We attribute the reduced saturation magnetization to the formation of non-ferromagnetic crystallization products. They may either be antiferromagnetic such as Fe-rich clusters, paramagnetic like fcc-Fe clusters or ferromagnetic with a very low (< 300 K) Curie temperature such as boron-rich Fe-B clusters. We have added an explanation of this observation to the manuscript.

(5) Reviewer 1: *In order to better clarify the structural origins of the Invar effect, I think the LAMMPS simulations are necessary here.*

Thank you for this suggestion. Unfortunately, the interatomic potentials for quaternary systems such as ours are not available. Therefore, there exists no base on which the simulations could be based. We nonetheless performed LAMMPS simulations in our research group in the past in order to understand the disordered structure and local arrangement of metallic glasses. We have added a comment on the use molecular dynamics (when applicable) as an alternative approach to the manuscript.

(6) Reviewer 2: *Writing about Invar effect in these bulk metallic glassy alloys one shall keep in mind that the CTE of metallic glasses based on iron is only 1.5-2 times lower than the CTE value for pure iron. Compare it with 1.5×10^{-6} K for Invar which is an order of magnitude lower than that of pure Fe. However, it is indeed lower than that of the base component which makes it different from other BMGs [Scripta Materialia, 58, (2008), 1106-1109]. One can also mention Cu-Zr based alloys CTE of which ($\sim 13 \times 10^{-6}$ K) is close to the average of CTE found for Cu and Zr.*

Thank you for pointing out this comparison of the Invar alloy with the studied BMGs. We have decided this is valuable as context for understanding the Invar effect in crystalline and amorphous metals and thus added it to the Introduction.

(7) Reviewer 2: *Did not the authors perform the same procedure for a crystallized BMG sample? One could have monitored the increase in the interatomic distances using Bragg peaks and compare with the BMG case.*

Thank you for raising this concern. We initially did not perform synchrotron in-situ XRD on the crystallized samples because we already saw the absence of the Invar effect in the dilatometry measurements. We have added the macroscopic dilatometry results of the crystallized samples to the supplementary materials. Additionally, we recently performed in-situ XRD on the crystallized samples at Diamond Light Source and could also confirm the

absence of the Invar effect at the atomic scale. These results of these experiments have also been added to the supplementary materials.

(8) Reviewer 2: *Main complaint is that the manuscript does not contain a clear explanation of the nature of the observed effect. It shall be stated more clearly in the discussion section.*

We appreciate this comment. We, however, would like to stress that we do provide the explanation in the text. With this work we show that the Invar effect is not just a macroscopic effect, but it has atomistic origins which can be observed through XRD. Through the choice of alloys and our investigation of the rPDF, we can attribute the atomistic Invar effect to Fe–Fe pairs. Nevertheless, the magnetically induced counteraction to the thermal expansion is not just seen in nearest-neighbor distances but is also observable in higher order atomic shells (and finally results in the macroscopically observed Invar effect). We therefore describe how the Invar effect manifests itself at the atomic scale which can serve as a reference for any model calculations or simulations that are made for ferromagnetic metallic glasses. Having experimental validation of how the Invar effect is working at the atomic scale removes the gap (of six to nine orders of magnitude) that any atomic model will have to bridge. This being said, we recognize that the explanations we provided in the text may appear unclear. We have now reworked the discussion section to more clearly state the new insights into the Invar effect.

(9) Reviewer 2: *I would recommend to perform ab-initio MD simulation of the atomic cells at several temperature values like it was done in [Appl. Phys. Lett. 93, 113104 (2008) and JOURNAL OF APPLIED PHYSICS 110, 043519 (2011)], for example. Contribution of the magnetic effects was studied in [PHYSICAL REVIEW B 68, 014407, 2003].*

Thank you for this suggestion. One problem with simulations of any sort is verifying the uniqueness of the solution and finding a simulated structure that reproduces the experimental observations. Especially amorphous alloys, and in particular those that are more complex than binary compositions, are difficult to model accurately. In this work we provide experimental evidence for the atomistic origins of the Invar effect in Fe-based metallic glasses.

(10) Reviewer 2: *One phrase should be reconsidered: - "Expansion of the average atomic distances..." can distances expand?*

We agree and have reworded the caption of Figure 8 to "Increase of the average atomic distances...".

(11) Reviewer 3: *The definition of the rate of volume expansion must be wrong, since at $T=T_0$, the equation is not equal.*

Thank you very much for pointing out this mistake. We have corrected the formula.

(12) Reviewer 3: *In Figure 2b, why T_g is not observed in the curve of the 1st peak?*

Thank you for raising this question. The glass transition is only seen very weakly in the first diffraction peak as transformations at the glass transition happen mostly at the SRO and are thus much more apparent in higher order diffraction peaks. Nevertheless, it is still detectable. Following the comment of the reviewer we added the arrow in Figure 2 to mark it.

(13) Reviewer 3: *In general, the first sharp diffraction peak around 3.0 Å⁻¹ corresponds to short-range order, however, the authors ascribed the shift of this peak (In Figure 2) to medium-range order. Then, what is the length scale of the so-called MRO?*

We thank the reviewer for raising this question. The first diffraction peak (low q) corresponds mostly to large distances (high r) via the Ehrenfest equation $r \sim \frac{1}{q}$. Higher order diffraction peaks are responsible for resolving finer features in real-space. The influence of the diffraction peaks on the pair-distribution function is studied in [Scudino, S. *et al.*, J. Alloys Compd. 639, 465–469 (2015)].

As far as length scales in real-space are concerned, we consider the first atomic shell (up to 3.5 Å) to be SRO, the next few atomic shells to be MRO (3.5 Å to 10 – 15 Å), and all correlations above 15 Å to be LRO (which is not present in BMGs).

(14) Reviewer 3: *The equation to calculate the coordination number may be wrong with the '+' in the integral.*

Thank you for pointing our attention to this equation. The coordination number has two contributing parts, the parabolic base level associated with the bulk density and the oscillating part related to the atomic shells. We have confirmed that the equation is correct, but we also understand that the summation within the integral may look odd. Therefore, we decided to include parenthesis to make clear that the sum of the two terms has to be integrated.

(15) Reviewer 3: *In calculating the weight factor of X-ray scattering, f_i is used with different meaning from the previous description. f_p is actually the X-ray scattering factor of element p at $q=0$ Å⁻¹.*

Thank you very much for pointing out this difference in meanings. We have adjusted the text and Figure 6 accordingly.

(16) Reviewer 3: *In the calculation of $\langle r \rangle$, I think the integral limits are not correctly set. $r_0 \sim r_1$ should spans the first coordination shell.*

Thank you for raising this concern. We have performed this analysis with several limits, integrands, baselines (because the probability-distribution-like part must be non-negative). The results always provide the same conclusion in the sense that the change in thermal expansion at the Curie temperature is always visible in all shells and most pronounced in the first one. Initiated by your comment, we have decided to choose $\frac{G(r)}{4\pi\rho_{\text{at}}r} + 1$ as the probability-distribution-like factor as this represents the probability (multiplied by the number of atoms) to find an atom at distance r around an arbitrary reference atom. We believe that this definition carries the most physical meaning. Furthermore, we have changed the integration limits to the limiting minima of the atomic shells. Moreover, we have found the absolute values of $\langle r \rangle$ to vary linearly with the expansion of the integration limits up to at least 10% variation with a very small proportionality constant. A 10% variation (be it through experimental errors or variation due to temperature) is by far more than what is reasonable to assume. In reality variation of 1 – 2% are more reasonable, which limits the influence of the integration limits on $\langle r \rangle$ to 0.6%. The definition and Figure 8 have been adjusted accordingly.

(17) Reviewer 3: “ *It is known that in crystalline materials the Invar effect can be clearly attributed to Fe10 atoms, but for metallic glasses we find that it is also visible in the full atomic shell, which comprises all atomic species.*” I remind the authors that some investigations focus on the chemical short-range order around Fe atoms in Fe-Ni crystalline Invar alloys.

Thank you for your comment. The Invar effect is rare in crystalline alloys. It seems to be that a particular local atomic arrangement is necessary for the Invar effect to manifest itself. For the crystalline Invar alloys the necessary Fe(-Fe) arrangement is only achieved if the correct amount Ni is available (locally). However, it was found that Ni does not show any exceptional volume dependency of its magnetic properties unlike Fe [Van Schilfgaarde, M. *et al.*, Nature 400, 46–49 (1999)]. Furthermore, EXAFS on crystalline FeNi Invar alloy also found thermal expansion around Fe to be exceptionally low while the same has not been observed for Ni [Yokoyama, T. , Eguchi, K. , Pys. Rev. Lett. 107, 1–4 (2011)]. We have added additional context to this statement to make it clearer.

Yours sincerely,
on behalf of all authors,

Alexander Firlus
Department of Materials
ETH Zurich

REVIEWERS' COMMENTS

Reviewer #1 (Remarks to the Author):

All of my questions have been concerns have been addressed in the revisions. The paper can be accepted as it is.

Reviewer #2 (Remarks to the Author):

The paper can be accepted for publication in its present shape.

Reviewer #3 (Remarks to the Author):

This paper contains systematic experimental data on the micro mechanism of invar effect in Fe based alloys, which provides a good basis for future theoretical analysis. The authors carefully and reasonably revised the reviewers' comments, which improved the quality of the article. I am very satisfied with this and suggest that it be published.